# Low-Temperature Soldering of Surface Mount Devices on Screen-Printed Silver Tracks on Fabrics for Flexible Textile Hybrid Electronics

**DOI:** 10.3390/s22155766

**Published:** 2022-08-02

**Authors:** Rocío Silvestre, Raúl Llinares Llopis, Laura Contat Rodrigo, Víctor Serrano Martínez, Josué Ferri, Eduardo Garcia-Breijo

**Affiliations:** 1Textile Research Institute (AITEX), 03801 Alicante, Spain; rsilvestre@aitex.es (R.S.); vserrano@aitex.es (V.S.M.); josue.ferri@aitex.es (J.F.); 2Departamento de Comunicaciones, Universitat Politècnica de València, 03801 Alcoy, Spain; rllinares@dcom.upv.es; 3Instituto Interuniversitario de Investigación de Reconocimiento Molecular y Desarrollo Tecnológico (IDM), Universitat Politècnica de València, Universitat de València, 46022 Valencia, Spain; lcontat@ter.upv.es

**Keywords:** e-textile, flexible hybrid electronics, soldering, screen printing

## Abstract

The combination of flexible-printed substrates and conventional electronics leads to flexible hybrid electronics. When fabrics are used as flexible substrates, two kinds of problems arise. The first type is related to the printing of the tracks of the corresponding circuit. The second one concerns the incorporation of conventional electronic devices, such as integrated circuits, on the textile substrate. Regarding the printing of tracks, this work studies the optimal design parameters of screen-printed silver tracks on textiles focused on printing an electronic circuit on a textile substrate. Several patterns of different widths and gaps between tracks were tested in order to find the best design parameters for some footprint configurations. With respect to the incorporation of devices on textile substrates, the paper analyzes the soldering of surface mount devices on fabric substrates. Due to the substrate’s nature, low soldering temperatures must be used to avoid deformations or damage to the substrate caused by the higher temperatures used in conventional soldering. Several solder pastes used for low-temperature soldering are analyzed in terms of joint resistance and shear force application. The results obtained are satisfactory, demonstrating the viability of using flexible hybrid electronics with fabrics. As a practical result, a simple single-layer circuit was implemented to check the results of the research.

## 1. Introduction

Printed electronics is a recent technology that permits the manufacture of electronic circuits on a wide variety of substrates. Among the materials used as substrates, the electronic circuits developed with flexible substrates, such as textile substrates [1,2], have particularly attracted the interest of the scientific community. Various applications involving textile printed electronics have been developed in different research fields. Basic patterns with conductive materials have been proposed to evaluate their operation as data or as energy transmission tracks with single-layer applications [3], such as RFID antennas [4], or multilayer applications [5,6], permitting, for instance, shielding [7] and better noise insulation [8].

Printing technologies have evolved, providing a gradual improvement of the resulting printed circuits. The technologies for printing, such as screen printing [9,10], inkjet printing [11], or roll-to-roll printing [12], including flexography [13,14], gravure [15,16], or offset [17], have adapted to the needs of this kind of technology.

Regarding the employed materials, great progress has also been made in the development of functional inks. They incorporate new materials ensuring, at the same time, adequate viscosities, and suitability. This allows them to be printed on a wide variety of printing equipment and substrates. Nanoparticle-based inks of many conductive materials have been developed, such as Ag [18,19], Au [20,21], Cu [11,22,23], Ni [24,25] or Pd [26], as well as different types of alloys [27].

The conductive tracks printed with the mentioned materials can be combined with surface mount electronic devices (SMD) giving rise to the so-called flexible hybrid electronics. Flexible hybrid circuits have had a great development on plastic substrates, but this kind of electronics using textile substrates presents some issues to be addressed. On the one hand, those relative to the printing of the conductive tracks on the textile substrate and, on the other hand, those concerning the incorporation of the SMD to these conductive tracks by soldering. Moreover, there exist problems that are intrinsic to the textile itself, such as washability, behavior under environmental changes (temperature and humidity), or bending that must be considered as well [28,29,30,31].

Regarding the printing of conductive tracks on textile substrates, the first issue found is related to the lack of specific inks for this kind of substrate. This means that it is necessary to check the most compatible ink with the textile substrate used. Moreover, not all textiles are likely to be used as substrates, since parameters such as thread material, thread diameter, roughness, or type of weave can make printing impossible [32]. In connection with this problem, difficulties may arise with the track size and the thickness of the layer deposited which, in addition to the print settings, will depend largely on the roughness of the fabric used.

The number of layers to be printed must also be considered, since usually, and due to the complexity of the circuits, the designs are multilayer. In this case, it is necessary to print conductor and dielectric layers, alternately, until the multilayer design is achieved. With textile substrates, short circuits between conductive layers are highly probable due to the difficulty of ensuring a good printing of the dielectric layers [32]. Only when the target circuit has a minimum number of components and is monolayer or bilayer, it should be printed on a textile substrate. For instance, in [33], an analog front-end was incorporated into the textile as the amplifier circuit of an electrocardiogram electrode. In the case of more complex circuits, flexible hybrid electronics in plastic substrates can be used, providing an alternative solution. In this case, the complex circuit is implemented in a flexible substrate, such as Kapton, and then, incorporated into the textile. For these kinds of solutions, problems arise when connecting the flexible hybrid circuit and the textile substrate, and the type of soldering between both systems is particularly important [34].

Regarding SMD, printing issues are found when using textile substrates, related to fit pad size limits when printing [35]. However, the most important problem arises with the soldering of the components and the pads printed on the textile, due to the high temperatures usually used in solder pastes. These high temperatures of soldering can deform, or even destroy, the textile substrate. Solder pastes can be replaced by conductive adhesives, but in the case of textile circuits, conductive adhesives do not usually provide good soldering results [36,37,38]. Thus, solder pastes ensure good conductivity and adequate fixation to the substrate [39], but their melting point must be compatible with the textile substrate [29,40].

The contribution of this paper is double. On the one hand, the paper determines the optimal design parameters (pattern classes) of screen-printed silver tracks on textiles, focused on printing an electronic circuit on a textile substrate. Even though the printing of the tracks has been successfully achieved in several works [3,41], it is necessary to study the results in depth when printing the substrate to incorporate an SMD. On the other hand, the paper carries out a study of the soldering of components. Several low-temperature soldering techniques and materials have been employed to ensure a satisfactory result in terms of electrical conductivity and soldering joint resistance. Finally, the paper studies the compatibility of different low-temperature solders with an SMD and with silver tracks.

## 2. Materials and Methods

The mechanical behavior of fabric can be characterized by parameters such as resistance or deformation. In electronic soldering, it is crucial to study the deformation of the fabric when applying temperature. For instance, a shrinkage of the fabric will reduce the size of the solder pads and may resist soldering or produce conductivity failures in it.

Natural fibers such as cotton, linen, silk, or wool have a burning point ranging from 148 °C for silk and up to 570 °C for wool [42]. Manufactured fibers present distinct temperature points for different points of degradation. The most commonly considered temperature points are, in ascending order, softening (Ts), melting (Tm), and burning points (Tb). For example, Nylon-6 has a Ts of 171 °C, a Tm of 220 °C, and a Tb of 450 °C [43]. In other common textile fibers such as acrylic, the values are 215 °C for Ts, 250 °C for Tm, and 290 °C for Tb, or for elastane, 180 °C for Ts, 230 °C for Tm, and 290 °C for Tb [43]. The temperatures where the most common manufactured materials begin to deform (Ts) range between 150 °C and 200 °C, although other materials such as polyethylene (PE) or polypropylene (PP) can deteriorate with TS below 140 °C. In general, these temperatures have an Sn-Pb eutectic melting-point temperature value close to 191 °C; therefore, it is necessary to use low-temperature solders.

In addition, below the Ts, mechanical deformations may occur. In polyester PET, with a Ts of 238 °C, mechanical deformations begin to occur from 140 °C [44], in cotton at 200 °C, in ramie at 250 °C, and in wool at 150 °C [45]. Above these temperatures, the shape and size of the fiber will change and shrink, and it will not return to its original state after cooling, which is called fiber thermal shrinkage. The percentage of length before heat shrinkage and after heat shrinkage is called fabric-shrinkage percentage. Shrinkage percentage also depends on the fabric constructions and the fabric quality. In general, natural fabrics have a shrinkage percentage of between 2 and 10%, whereas manufactured fabrics have a shrinkage percentage of between 0 and 8%. To avoid any mechanical deformation, a soldering temperature of less than 200 °C should be set. These soldering temperatures are only possible with the use of low-temperature solders.

The preeminent solder in the area of electronics has been based on Pb. Sn-Pb eutectic solder has been used for decades due to its excellent properties, good wettability, high reliabilities, and low cost [46]. The RoHS (Restriction of Hazardous Substances) and WEEE (Waste Electrical and Electronical Equipment) directives have made Pb-based solders disappear, giving rise to a new type of Pb-free solder [47]. In recent years, different alloys have been developed to obtain a solder with characteristics similar to Sn-Pb. Among the different alloys obtained, the so-called low-temperature alloys stand out.

Low-temperature solders permit less thermal deformation of the components, increasing the quality of the solder joint due to a lower deformation of the components. In addition, they provide a reduction in costs due to the materials used and the energy-cost saving [40].

In general, low-temperature solders are based on Sn or In alloys. Depending on these two elements, three groups can be established: Sn-Bi based, Sn-Zn based, and In (In-X) based.

Sn-Bi solder has good properties due to its low melting point. 42Sn-58Bi eutectic has a melting point of 138 °C. Other advantages are its low cost, due to the lower cost of Bi [40], the reduction of the generation of bubbles, and meeting the requirements of low-temperature air tightness [48]. However, this alloy has some problems, such as poor brittleness and wettability [40].In-based solder has a very low melting point, good conductivity, good thermal conductivity as well as, fatigue resistance, ductility, and wettability. 52In-48Sn eutectic has a melting point of 118 °C. However, its cost is appreciably higher, due to the high cost of In [49,50].Sn-Zn lead-free solder has a low melting point. 91.2Sn-8.8Zn eutectic has a melting point of 199 °C. However, this alloy has some problems such as poor wettability, coarse structure, and low oxidation resistance [51].Sn-Bi-based solder alloys have a low melting point solder and reliable properties. Sn-58Bi with a melting point of 138 °C is a representative solder with low cost, good solderability, and environmentally friendly characteristics [52].

To improve the properties of the different alloys, additional alloying elements can be incorporated. Jiang, N et al. [53] reviewed different alloys based on Sn-Bi, reporting more than 40 alloys, including alloy 50Sn-35Bi-12I, which had a melting point of 100 °C and was the one with the lowest temperature obtained. Table A1 shows some of the main commercial solder pastes between 100 °C and 200 °C, their alloys, and their corresponding liquidus and solidus temperatures. Liquidus is the lowest temperature at which an alloy is completely liquid; solidus is the highest temperature at which an alloy is completely solid. When liquidus and solidus are at the same temperatures, the term eutectic is usually used, which is the lowest melting point of the system.

### 2.1. Textile Substrate

The fabric used for the study is a 100% waterproof cotton substrate with the characteristics shown in Table 1 and Table 2. This fabric has been chosen for obtaining excellent results in a previous study of screen printing silver conductors on different textile substrates [32].

A shrinkage study was carried out to determine the percentage of shrinkage of this fabric and provide a treatment that allows a 0% shrinkage for the thermal treatment of curing the silver ink and soldering. In the first process, the fabric was subjected to washing with neutral soap using six resistant cotton programs at temperatures of 90, 60, 50, 40, 30 °C, and a cold wash, with a spin cycle at 1200 rpm in all programs. The washing machine used was the model WM12E467EE (Siemens Aktiengesellschaft, Munich, Germany). In the second process, the fabric was subjected to steam ironing at 215 °C using the G9222F0 model (Rowenta Irons, Millville, NJ, USA). The third process was to introduce it into an oven FED-115 (Binder GmbH, Tuttlingen, Germany) at 130 °C for 30 min, using the same curing cycle as conductive silver inks. Finally, it was subjected to an oven temperature of 200 °C for 3 min, higher than 191 °C which is the melting point temperature of the Sn-Pb eutectic, using the same temperature cycle as the one used in a typical solder paste.

### 2.2. Conductive Tracks

When an electronic circuit needs to be printed, it needs to be as dimensionally stable as possible. Therefore, it is not necessary to use shrinkable inks, these can be used to join different circuits in a textile [54]. The most important feature, in this case, is a resulting high conductivity to reduce the resistance of the tracks. Although there are no specific conductive inks to apply on textiles, different conductive inks were selected based on their high-line-resolution property for printing on flexible substrates. The conductive inks used (Table 3) are made of silver with different characteristics of solid contents, viscosities, flexibility, and stretchability. All of them were specific inks for screen printing: SCAG-003, SCAG-004, and SCAG-005 (Mateprincs, Navarra, Spain).

#### Minimum Design Parameters

In printed electronic circuits, it is necessary to determine the minimum sizes for the track (conductor) and the gap (insulation) to make electronic circuits on any substrate. These two parameters determine the so-called pattern classes. The track width (TW) and the gap between conductors, namely track-to-track (TT), track-to-pad (TP), and pad-to-pad (PP) gaps need to be defined.

To carry out a study of these parameters using the aforementioned three inks, a specific pattern (Figure 1) with a series of tracks and SMD land patterns was designed.

The designed pattern consists of tracks of different sizes and orientations, as well as a circle and a serpentine that permitted the investigation of the printing that resulted. On the other hand, a series of SMD land patterns were included to assess the effect of printing on its dimensionality. The dimensions of the land patterns were based on the recommendations of the standard IPC-SM-782. For these land patterns, the minimum track width (TW) was 10 mils (0.254 mm), and the minimum pad-to-pad gap (PP) was 8 mils (0.203 mm).

Groups of four tracks were drawn, in the direction of the weft (0°), warp (90°), and at 45° and 315°. The widths of these tracks (TW) were 12, 15, 20, 25, 30, 40, and 50 mils (0.304, 0.381, 0.508, 0.635, 0.762, 1.016, and 1.27 mm). The gaps between tracks (TT) in each group were 12, 15, 20, 25, 30, 40, and 50 mils (0.304, 0.381, 0.508, 0.635, 0.762, 1.016, and 1.27 mm), respectively. In addition, four tracks of 80 mils (2.032 mm) with a length of 3937 mils (100 mm) were drawn.

In some of the SMD land patterns, tracks were drawn between pads, in this case, the width used was 10 mils (0.254 mm). Finally, two serpentines at 0° and 90°, and a ring with a width of 30 mils (0.762 mm) were also drawn.

In order to print the pattern, the manufacturing technology used was serigraphic technology. To check the effect of the thickness and resolution of the conductor layer, three types of mesh were used for the screens, namely PET 1500 90/230-48 PW, PET 1500 130/330-34 PW, and PET 1500 150/380-34 PW (Sefar AG, Heiden, Switzerland) with 90, 130, and 150 threads/cm, respectively. The main features of the mesh are shown in Table A2. Afterwards, to transfer the pattern to the screen mesh, a UV film Dirasol 132 (Fujifilm, Tokyo, Japan) was used. The patterns were transferred to the screen by using a UV light source unit IC-5000 (BCB, Pamplona, Spain).

Printing was carried out using an E2XL from an EKRA screen printer (ASYS Group, Dornstadt, Germany) with a shore 75° hardness squeegee, 60° squeegee angle, 1 mm snapoff, 3.5 bar force, and 100 mm/s. After the deposition of the inks, these were cured in an air oven FED-115 from BINDER at 130 °C for 15 min to use the same curing characteristics for all the inks.

One parameter that affects the thickness and width of the track is the number of printed layers. Normally a single print is made, but it is usually insufficient in textiles since, in this case, the tracks do not conduct due to the rough surface of the textile itself that makes printing correctly difficult. J. Ferri et al. [32] demonstrated the necessity of making several impressions to obtain an adequate resistance depending on the textile substrate. For this reason, two, three, and four impressions have been made in this work. The printings were made one after the other and before the thermal curing.

The dimensional characterization was carried out with a magnifying glass AM9915MZT from Dino-Lite (AnMo Electronics Co., New Taipei City, Taiwan). Resistance measurements and electrical continuity were made with a Fluke 8845A multimeter (Fluke Co., Everett, WA, USA). The resistance was measured using the four-probe method. Data were sent to a PC via RS232 connection, they were stored and analyzed with FlukeView^®^ software (version 3.8.304, Everett, WA, USA).

### 2.3. Soldering Methods and Test Pattern: Low-Temperature Solders

The solder pastes used in this study are commercially available. Indium NC-SMQ80 Indalloy #1E of Sn-In alloy and Indium5-LT Indalloy#281 of Sn-Bi alloy (Indium Corporation, Clinton, NY, USA), TS391lT50 (ChipQuick, Hamilton, ON, Canada) of Sn-Bi-Ag alloy, and SSLTNC (SRA Soldering Products, Walpole, MA, USA) of Sn-Bi-Ag alloy were used (Table 4).

In order to check the soldering of the four solder pastes, different tests with a second pattern (Figure 2) were designed. This pattern consists of the four basic SMD land patterns (Table A3): 0805, 1206, SOT23, and SOIC8 with their corresponding tracks. These tracks allow the checking of the component’s electrical characteristics during the tests.

On the 0805 land pattern, 10 kΩ resistors (CRGP0805F10K) were soldered; on 1206, LED diodes (LTST-C150CKT), on SOT23, double diodes (TPS7A0533PDBZR), and finally, on SOIC8, operational amplifiers (NCV4949CDR2G) were used.

Solder paste printing was carried out using stencil printing. The screen printer was an E2XL from the EKRA (ASYS Group, Dornstadt, Germany) with a stainless steel stencil of 150 µm, 90° squeegee angle, 0 mm snap-off, 2.5 bar force, and 50 mm/s. Then, components were placed using a pick and place ProtoPlace S (LPKF Laser & Electronics AG, Garbsen, Germany).

#### 2.3.1. Soldering Profiles

The continuous five-zone reflow oven AE-R330 (SMT MAX, Chino, CA, USA) was used for soldering the samples.

The profiles supplied by the ink manufacturers were based on their use with PCBs, so the profile had to be modified to suit textiles. For the experiment, the soldering profiles were based on datasheets by recommendation of solder paste manufacturers and were adjusted until a correct soldering was obtained. The measurements of the temperature of the different profiles were obtained using a thermocouple data logger USB TC-08 (PICO technology, Saint Neots, UK).

#### 2.3.2. Soldering Joint Resistance

The joint electrical resistance among the pad of the component, the soldering, and the land pattern, can give an idea of a good soldering. Resistance measurements were made with a Fluke 8845A multimeter (Fluke Co., Everett, WA, USA). The soldering joint resistance was measured using the four-probe method. Data were sent to a PC via RS232 connection and they were stored and analyzed with FlukeView^®^ software (version 3.8.304, Everett, WA, USA).

#### 2.3.3. Shear Test

The shear test method is designed to test and evaluate the endurance of the solder joint between component terminals and lands on a substrate, using a shear-type mechanical stress (standard IEC 62137-1-2:2007). 

To measure the shear strength of welds, a V-275.431 PIMag^®^ voice coil linear actuator with force sensor from Physik Instrumente (PI) GmbH & Co. KG (Karlsruhe, Germany) was used.

Figure A1 shows the experimental set up used for X force and for acquiring data during the soldering characterization. The electrical measurement taken was made with a Fluke 8845A multimeter (Fluke Co., Everett, WA, USA). The 10 kΩ resistance measurement was performed in resistance mode and using the four-probe method. The measurements of the LED diode and the double diode were carried out in diode mode, measuring its forward voltage and, finally, the operational amplifier was checked by measuring the saturation voltage at its output.

#### 2.3.4. Optical Microscopy

All optical characterization was carried out with a magnifying glass AM9915MZT from Dino-Lite (AnMo Electronics Co., New Taipei City, Taiwan).

## 3. Results and Discussion

### 3.1. Shrinkage Percentage

Figure 3 shows the shrinkage percentage after the washing and ironing (B) and curing processes (C and D) explained in Section 2.1. Shrinkage in the werf of approximately 2.5% was observed for all washing programs, after washing and ironing. Warp shrinkage was higher at about 5.5%. In general, the warp yarns are under more strain due to interlacement than the weft yarns. After curing at 130 °C, the percentage was reduced to a range between 0.5 and 1.5% in both, weft and warp, but in the case of washing at 90 °C, the percentage was reduced to 0%. After curing at 200 °C, a reduction of 0% is obtained for all washing programs. In conclusion, the most critical process was washing and ironing, with a better result observed when washing at 90 °C.

Washing at 90 °C, ironing, and curing at 130 °C will be used as a previous treatment applied to the textile to obtain a substrate as stable as possible for the soldering stage.

### 3.2. Minimum Design Parameters

#### 3.2.1. Electrical Parameters

The most relevant data obtained from the study on pattern one are presented since the whole study exceeds the scope of this publication. The used nomenclature is the following: M3, M4, and M5 corresponding to the silver inks SCAG-003, SCAG-004, and SCAG-005, respectively; T12 and T15 for the tracks of 12 mils (0.305 mm) and 15 mils (0.381 mm), respectively.

To determine the correct printing of the pattern, an electrical continuity test was carried out using a Fluke 8845A multimeter. As a result, it can be noted that a high percentage of the results present open circuits (Figure 4a) or shorts circuits (Figure 4b).

Figure 5 shows the percentage of tracks with no errors for T12 (Figure 5a) and T15 (Figure 5b). The open circuits were mainly found with the printing of a mesh of 150 threads/cm with the three inks. This problem could be due to the lower open area of PET1500 150/380-34PW, 12%, versus 25% or 27% of the two other meshes, involving a lower ink deposition.

The percentage of shorts was low, with the worst case being lower than 30%, due to the excessive ink deposition because of the increase in the number of printings. Most of the shorts were found with four printing layers, with meshes of 90 and 130 threads/cm, and, particularly with M4.

Thus, from Figure 4, it can be deduced that the lower printing limit for this work was a track of 15 mils (0.381 mm), with three printing layers for M3 and M5 inks using a mesh of 90 threads/cm. In addition, the resulting tracks could be accepted with M3 ink with three printings of a mesh of 130 threads/cm.

As observed in Figure A2, for tracks with 20 mils (0.508 mm) or higher, all of the three inks achieved 100% of tracks with no errors for three layers with meshes of 90 and 130 threads/cm. A track of 100% with no errors was also obtained with M3 ink with three printings for a mesh of 130 threads/cm.

Once the best printing configurations were determined to ensure the physical integrity of the track, the influence of the printings on the track resistance was studied. Figure A3 shows the average resistance of the set of tracks from 15 to 80 mils (0.381 to 2.032 mm). It can be observed that a high resistance was obtained in tracks with a mesh of 150 threads/cm with any of the three inks. In general, M4 ink obtained higher resistance values. Removing the high resistance cases from the study, Figure 6 shows the results of the lower resistances obtained. In conclusion, it can be assessed that the best printing as a function of the resistance was obtained with M5 ink with four layers and a mesh of 90 threads/cm. However, if the percentage of tracks with no errors is considered, the best relationship between low resistance and tracks with no errors was obtained with M5 ink, three layers, and a mesh of 90 threads/cm.

Based on this unique printing configuration, a study of the value of the resistance as a function of the orientation was conducted. Figure 7 shows the average of the resistance of the set of tracks from 15 to 80 mils (0.381 to 2.032 mm) as a function of the printing orientation. A variation of the resistance depending on the orientation can be observed, lower in the case of tracks traced in the direction of weft (0°) and warp (90°). The highest resistance corresponds to the weft direction (0°) since it has a low density of threads/cm compared to the warp direction (90°) whose density is higher, and, therefore, its resistance is lower. The tracks printed at 45° and 315° have the least resistance since their net density is greater when combining the densities of the weft and warp. This difference between orientations decreases with the thickness of the tracks since the density has lower influence due to the greater contribution of ink.

The individual value of resistance can be seen in Figure 8. The deviation is higher for small values of the track width and is practically zero for tracks of 80 mils (2.032 mm). The 30 mil (0.762 mm) serpentines have a very similar deviation to 30 mil (0.762 mm) tracks. This effect may be due to the variation in the width of the track because of the printing. In the case of the ring, since there is only one ring, its deviation cannot be determined.

#### 3.2.2. Dimensional Parameters

A fundamental parameter in the development of electronic circuits is the resolution of the track width (TW) obtained in the printing. Several factors can influence this parameter: First, the type of ink due to the particle size it contains. Mateprincs does not give exact information about the particle size but indicates that it is less than 10 µm; Second, the mesh of the screen, since the greater number of the mesh, the higher the resolution obtained; Third, the number of printed layers, since the greater number of layers, the higher the width obtained.

From the results, the average variation of the track width depending on the orientation and for tracks between 15 and 80 mils (0.381 and 2.032 mm) is 6.14% for 0°, 1.09% for 45°, 6.47% for 90°, and 5.26% for 315°. These data coincide with the average resistance of the set of tracks from 15 to 80 mils (0.381 to 2.032 mm), shown in Figure 7, due to the density of the weft with respect to the warp.

As mentioned previously, the number of prints influences the resolution. The results obtained show that the track width of 15 mils (0.381 mm) printed at 0° with M5 ink and a mesh of 90 threads/cm, ranges between 17.74 ± 1.90 mils (0.45 ± 0.05 mm) with two printings, and 21.95 ± 1.78 mils (0.56 ± 0.05 mm) with four printings. The number of the mesh also influences the result, the track width of 15 mils (0.381 mm) printed at 0° with M5 ink and a mesh of 150 threads/cm, ranges between 15.64 ± 1.27 mils (80.39 ± 0.03 mm) with two printings, and 15.75 ± 2.76 mils (0.40 ± 0.07 mm) with four printings.

Table 5 shows the track width obtained for each track size based on the printing configuration, M5 ink, three layers, and a mesh of 90 threads/cm. From the table it can be deduced that the variation in the resistance per track is not totally related to the variation in the width of the track. As mentioned, there may be other factors that influence the variation in resistance, such as the density of the fabric in the weft and warp and also the amount of ink that is deposited in the crests and valleys of the fabric.

The variation in the width of the tracks implies changes in the land patterns, but this variation is not so significant that welding failures occur. Figure 9 shows the land pattern of a PLCC32 with pads of 25 × 80 mils (0.635 × 2.032 mm). The theoretical pads are shown in red and the real pads in blue. The variation is more accentuated in the warp direction (90°), approximately 6%, as aforementioned.

### 3.3. Soldering Test

#### 3.3.1. Soldering Profiles

The reflow solder oven used only allows for configuring the temperature in each of the five stages of the curing process. The temperature of each stage was adjusted until the temperature curve was as close as possible to the profile indicated by the solder paste instructions. During the adjustments, it was noted that the reflow point changed depending on the substrate. Hence, the reflow soldering profiles were tuned adapting the temperature values to the substrate used. The final profiles obtained are shown in Figure 10. The temperature peaks applied were 169 °C for SRA SSLTNC, 158 °C for Indium #1E, 160 °C for Indium #281, and 178 °C for Chipquick TS3922T50. The temperature peaks, as well as the time intervals where these peaks were applied, were slightly different from those recommended by the solder manufacturers (Table 4).

#### 3.3.2. Soldering Joint Resistance

The soldering joint resistance of the packages presented in Section 2.3 (0805, 1206, SOT23, and SOIC8 packages) was measured. The resulting contact resistances of the solder joints are shown in Figure 11. SRA SSLTNC solder paste showed the lowest resistance and the lowest dispersion of all packages. The 1206 package had the highest dispersion due to the physical difficulty of measuring the joint. This low strength in Indium indalloy #1E may be due to the inclusion of In in the alloy. In any case, all solders had a junction resistance below 10 mΩ. The soldering was reliable in all cases and 100% of the joints had correct electrical characteristics.

#### 3.3.3. Shear Test

Table 6 shows the maximum shear force applied on the *X* axis for three types of land pattern. Figure A4 shows the variation of resistance with increasing applied force, for a 10 kΩ resistor with a 0805 land pattern. The variation of the resistance was minimum for the four solder pastes up to an abrupt variation caused by the rupture of the solder. This minimal variation indicates that the solder joints maintain their integrity up to the maximum force they can withstand. In the case of land patterns with two pads, SRA SSLTNC and ChipQuick TS3922T50, solder pastes supported a much higher voltage than the other two solder pastes for the 0805 land pattern and a similar voltage for the 1206 land pattern. This difference is also present for the SOT23 land pattern, but with a smaller difference in strength. These differences may be due to the alloys of the solder pastes, the SSLTNC with 42Sn/57Bi/1Ag, and the TS3922T50 with 42Sn/57.6Bi/0.4Ag, both have a similar Bi content, and the one that supports most force is the one with the highest silver content. Indium Indalloy #281 has a similar concentration of Bi but no Ag, and lastly, Indium Indalloy #1E has an Sn/In alloy. Silver can benefit soldering with silver conductors.

#### 3.3.4. Optical Characterization

In general, the solder joint cannot be considered to be very good compared to similar solder joints on PCBs, but based on the results of shear-force tests, they can be considered acceptable. The observation of the optical magnifications (Figure 12, Figure 13, Figure 14 and Figure 15) shows an acceptable wetting of the SRA SSLTNC solder paste. The wetting of the Indium Indalloy #1E solder paste is poor, forming irregular shapes. In the Chipquick TS3911T50 and Indium Indalloy #281 solder pastes, voids and pores can be observed. These problems may be due to a low welding temperature or an absence of Ag in the alloys.

#### 3.3.5. Circuit on Textile

A simple oscillator circuit based on the IC555 was designed (Figure A5) and implemented (Figure 16) on a cotton textile substrate in order to check the reliability of the research. The circuit consisted of resistors, capacitors, an LED diode, an IC555, and a battery holder. The resistors, capacitors, and led used a 0805 package, whereas the IC555 used a SOIC8 package. The minimum track width was 20 mils (0.508 mm), and the minimum separation between tracks was 12 mils (0.305 mm). A finger-activated switch was also printed. The best printing and soldering option determined in the study was followed. A solder mask layer was also printed with dielectric ink (Mateprincs, Navarra, Spain) to facilitate soldering.

## 4. Conclusions

The direct realization of electronic circuits in textiles is not feasible in the case of complex circuits, with multilayer layouts and small track widths or minimum separation between tracks. In these cases, it is better to implement the circuit on a flexible PCB and incorporate it into the textile. However, there are basic circuits that may be interesting to implement directly onto textile fabrics, for instance, an amplifier as close to an electrode as possible. In this case, the layout can be a single layer, with less-demanding track sizes and widths. For those cases, it would be interesting to carry out a study to determine the best way to implement the circuit on a textile substrate.

The choice of the textile substrate is critical, and tests must be carried out to obtain a good printing of the conductors. Once the best textile substrate has been selected, previous treatments, such as washing, ironing, and drying, must be applied to avoid size modifications of the textile during the track printing and SMD soldering stages. Next, the minimum track width and separation between tracks must be established to be able to implement the circuit. The influence of the tracks on the resistance of each net must also be analyzed. This will be determined by the type of ink, the number of prints, and the type of mesh used. Lastly, the best low-temperature solder paste, compatible with the component and conductive trace, must be determined.

In this work, a 100% waterproof cotton substrate with the most compatible silver ink was used. The best printing option was achieved for three layers of conductive ink, printed with a mesh of 130 threads/cm. Under these conditions, the minimum track width obtained, as well as the separation between tracks, was 15 mils (0.381 mm). The best results were obtained with SRA SSLTNC solder paste, possibly due to its composition of In and Ag. The verification of this process was assessed with a simple single-layer circuit. 

## Figures and Tables

**Figure 1 sensors-22-05766-f001:**
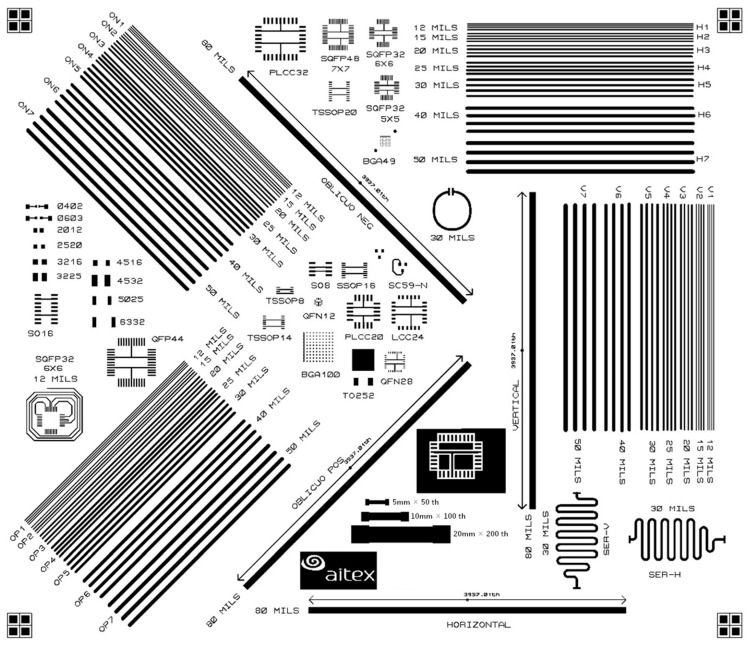
Patterns with tracks of different sizes and orientations, land patterns of different sizes, and pitches used to determine the minimum design parameters.

**Figure 2 sensors-22-05766-f002:**
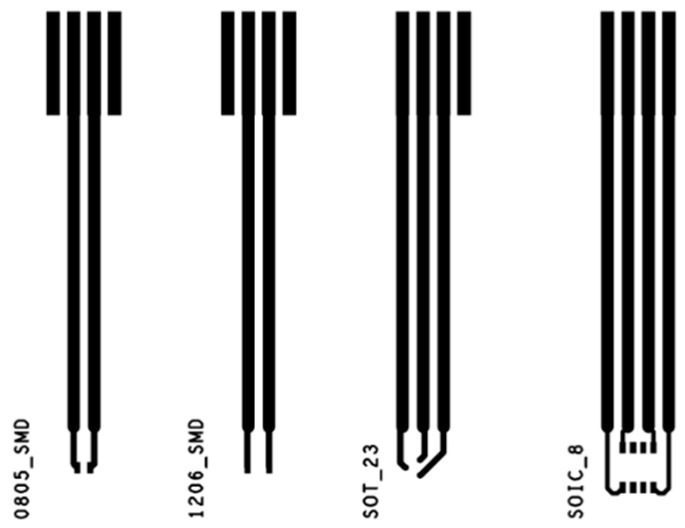
The pattern to check soldering of four basic SMD land pattern such as 0805, 1206, SOT23, and SOIC8.

**Figure 3 sensors-22-05766-f003:**
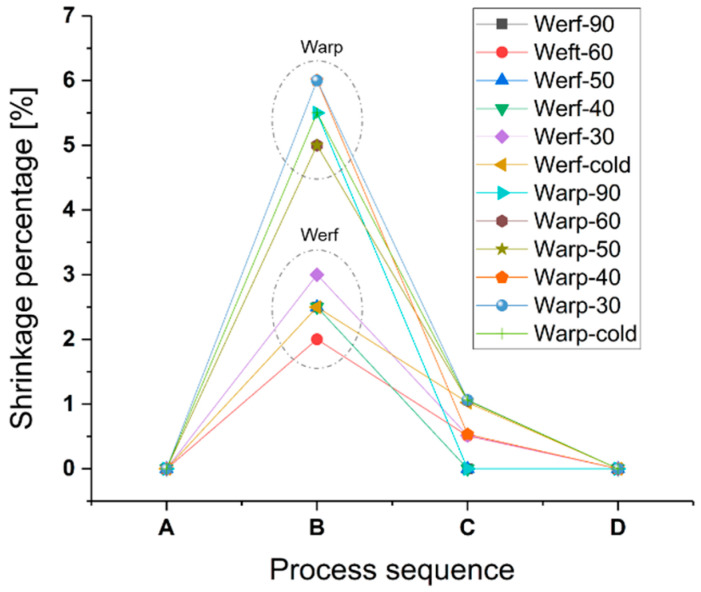
Shrinkage percentage for the process sequence: (A) without treatment, (B) washing and ironing, (C) oven at 130 °C/30 min, and (D) oven at 200 °C/3 min. Nomenclature: Werf/Warp-wash temperature.

**Figure 4 sensors-22-05766-f004:**
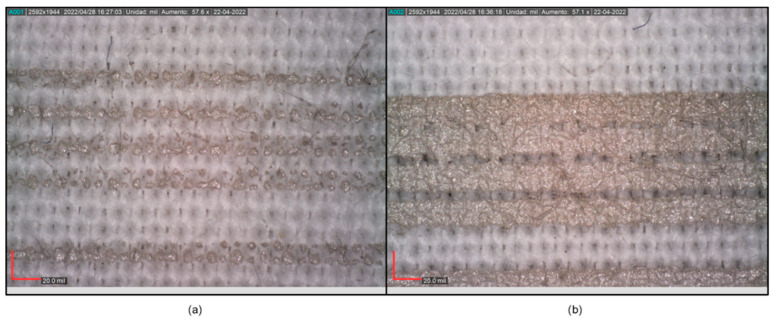
(**a**) Open circuit: 12 mil (0.305 mm) tracks printed using SCAG-005 ink with two printings using a mesh of 150 threads/cm; (**b**) short circuit: 12 mil (0.305 mm) tracks printed with SCAG-005 ink with four printings using a mesh of 130 threads/cm.

**Figure 5 sensors-22-05766-f005:**
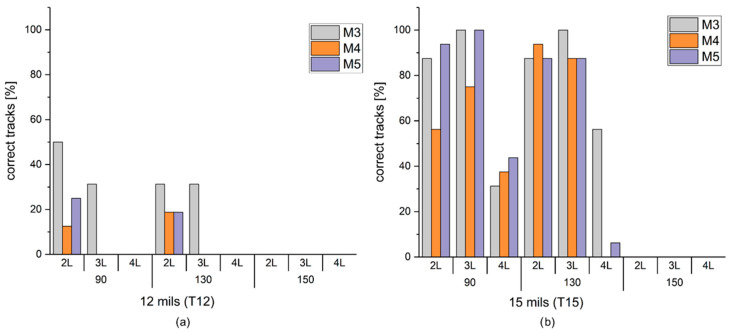
(**a**) Percentage of T12 tracks with no errors (correct tracks) as a function of inks (M3, M4, and M5), number of printings (2L, 3L, and 4L for two, three, and four layers) and mesh sizes (90, 130, and 150 threads/cm). (**b**) Percentage of T15 tracks with no errors (correct tracks) as a function of the same inks, number of printings, and mesh sizes.

**Figure 6 sensors-22-05766-f006:**
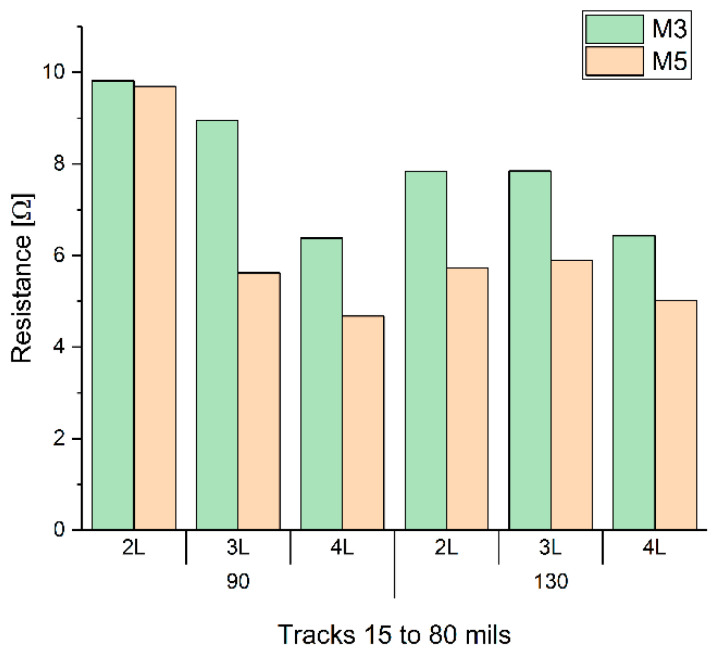
Average resistance of the set of tracks from 15 to 80 mils (0.381 to 2.032 mm), as a function of inks, M3 and M5, number of printings (2L, 3L, and 4L) and meshes of 90 and 130 threads/cm.

**Figure 7 sensors-22-05766-f007:**
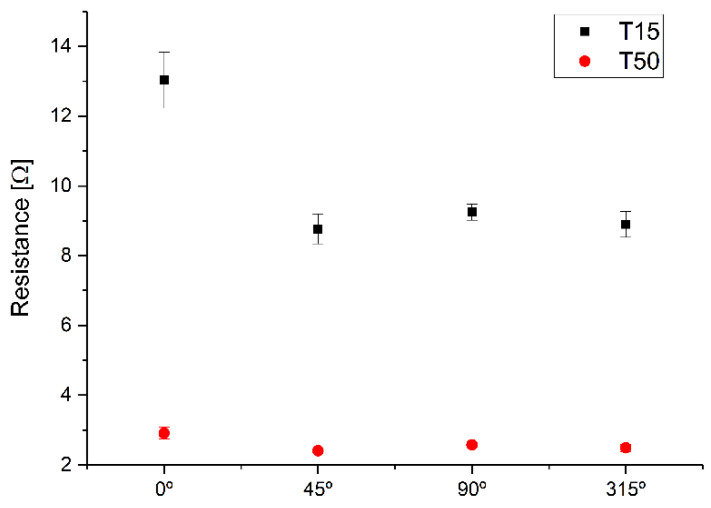
Average of the resistance of the set of tracks of 15 and 50 mils (0.381 and 1.27 mm) printed with M5 ink as a function of the printing orientation.

**Figure 8 sensors-22-05766-f008:**
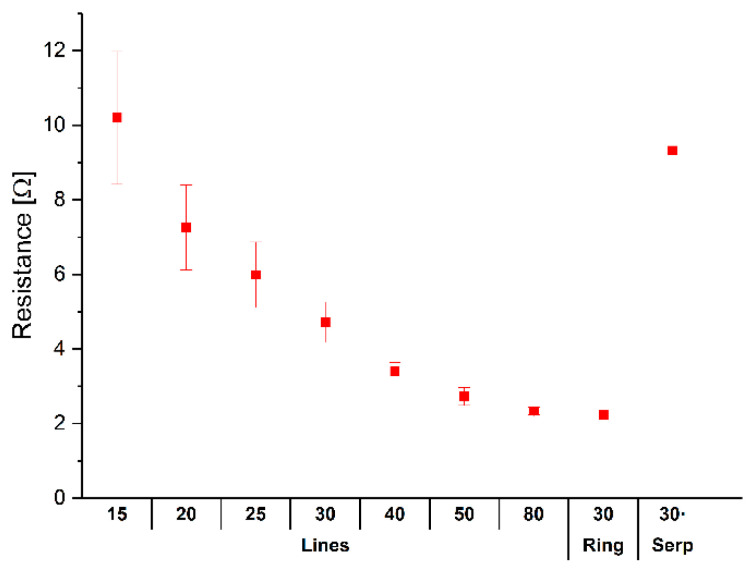
Resistance of each track from 15 to 80 mils (10.381 to 2.032 mm) printed with M5 ink, three layers, and 90 threads/cm mesh.

**Figure 9 sensors-22-05766-f009:**
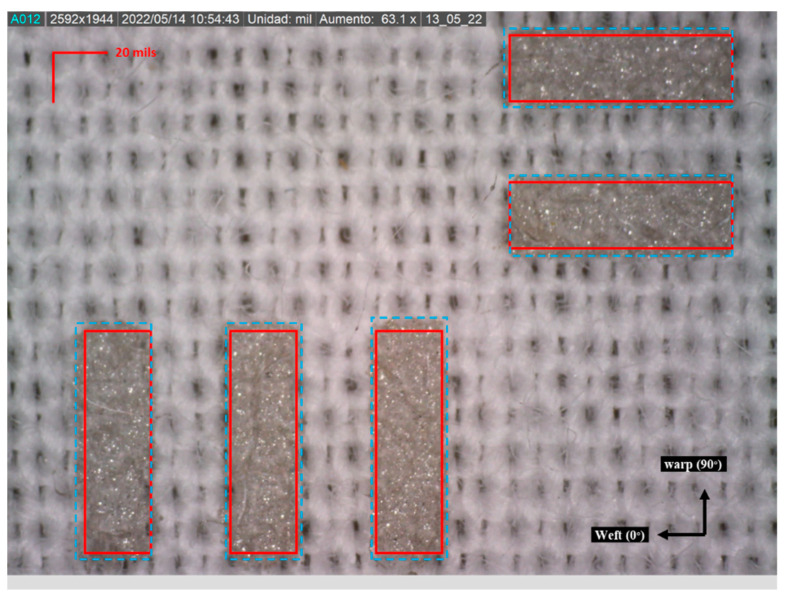
Variation in the land pattern of a PLCC32. The theoretical zones of the pads are represented in red and the real ones in blue.

**Figure 10 sensors-22-05766-f010:**
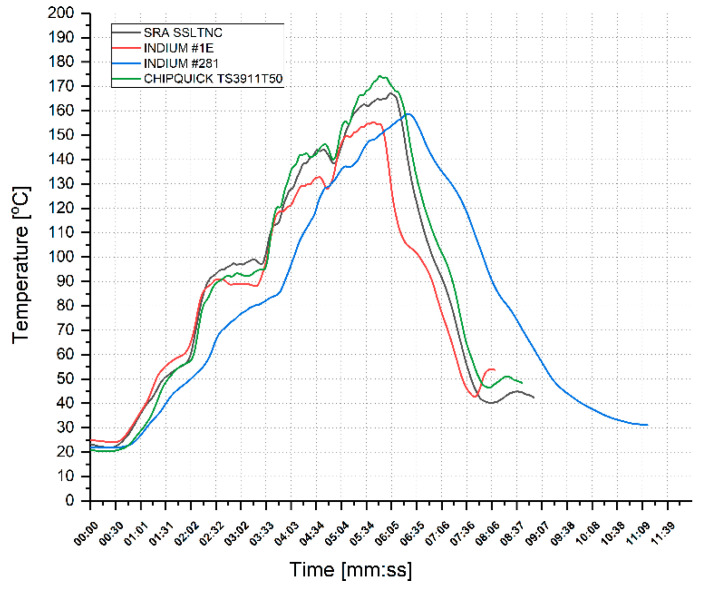
Temperature profiles used for soldering.

**Figure 11 sensors-22-05766-f011:**
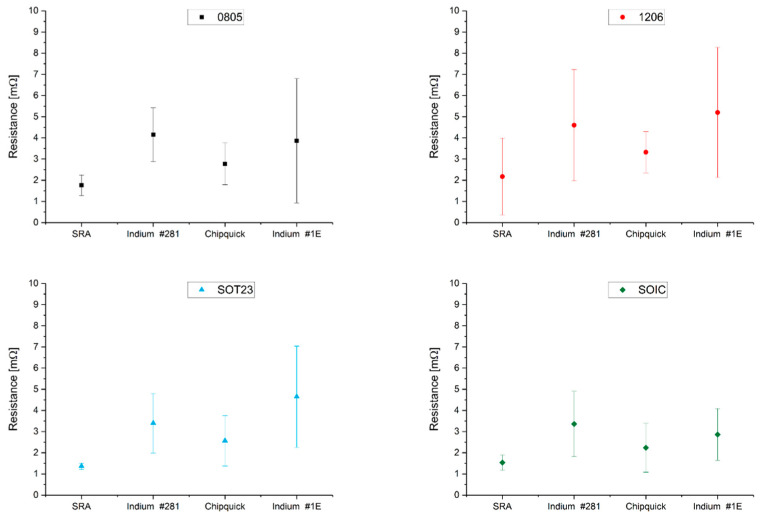
Solder joint resistance of 0805, 1206, SOT23, and SOIC8 packages. Units: mΩ.

**Figure 12 sensors-22-05766-f012:**
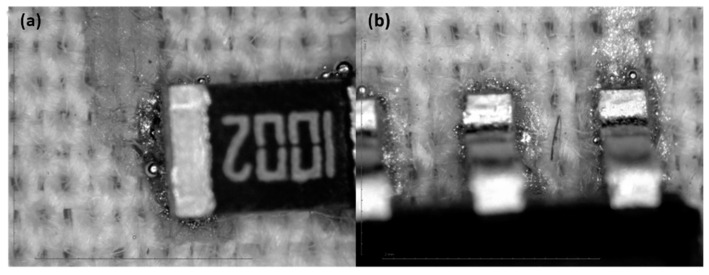
Soldering microscope pictures of 0805 (**a**) and SOIC8 (**b**) packages with solder paste Chipquick TS3911T50.

**Figure 13 sensors-22-05766-f013:**
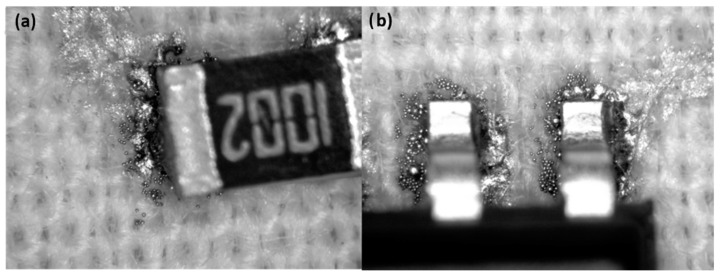
Soldering microscope pictures of 0805 (**a**) and SOIC8 (**b**) packages with solder paste Indium Indalloy #281.

**Figure 14 sensors-22-05766-f014:**
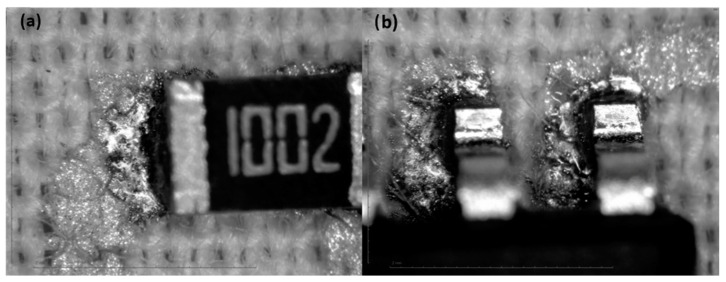
Soldering microscope pictures of 0805 (**a**) and SOIC8 (**b**) packages with solder paste Indium Indalloy #1E.

**Figure 15 sensors-22-05766-f015:**
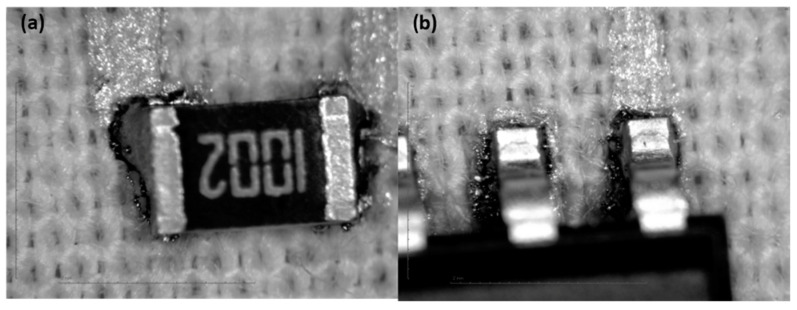
Soldering microscope pictures of 0805 (**a**) and SOIC8 (**b**) packages with solder paste SRA SSLTNC.

**Figure 16 sensors-22-05766-f016:**
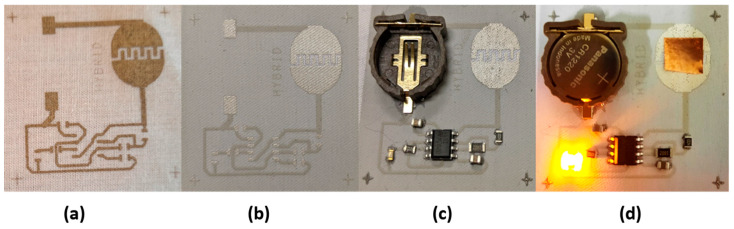
Image of oscillator circuit: (**a**) layout, (**b**) solder mask, (**c**) Toff oscillator time, and (**d**) Ton oscillator time.

**Table 1 sensors-22-05766-t001:** Fabric characteristics (I): composition and ligament.

Fabric	Picture	Weft Material	Warp Material	Ligament	
100% Waterproof Cotton	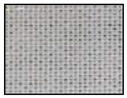	Cotton	Cotton	Taffeta	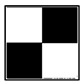

**Table 2 sensors-22-05766-t002:** Fabric characteristics (II): size and weight characteristics.

Fabric	Weft Density (Thread/cm)	Warp Density (Thread/cm)	Fabric Density (Thread/cm^2^)	Wire Weft Diameter (µm)	Wire Warp Diameter (µm)	Thickness (µm)	Grammage (g/m^2^)
100% Waterproof Cotton	32	44	76	160	160	130 ± 5	115 ± 6

**Table 3 sensors-22-05766-t003:** Silver inks characteristics.

	SCAG-003	SCAG-004	SCAG-005
Sheet Resistivity (Ω/sq@25 µm)	<0.05	<0.06	<0.02
Solids (%)	70–81.5	70–80	75–80
Density (g/cm^3^)	3.16	3.1	3.12
Viscosity (Pas)	14.3–14.8 @200 s^−1^	39.9 @10 s^−1^	40–50 @10 s^−1^
Screens polyester (Threads/cm). µm	130.30	650.14	130.30
Curing Box oven	130 °C—20–30 min	130 °C—20–30 min	160 °C—10 min
Properties	Silver/Silver-Copper Very high definition	Silver/Silver-Copper Very high definition	Silver High definition High Conductivity

**Table 4 sensors-22-05766-t004:** Solders used: main characteristics.

	Alloys	Liquidus	Solidus	Mesh Size	Metal Load	Peak Temperature Time Max.
Indium Indalloy #1E	48Sn/52In	118	118	T3	89%	143–163 °C <60 s
Indium Indalloy #281	42Sn/58Bi	138	138	T3	84%	163–183 °C <60 s
Chipquick TS3911T50	42Sn/57.6Bi/0.4Ag	139	138	T4	90%	165 °C <40 s
SRA SSLTNC	42Sn/57Bi/1Ag	140	139	T3	87–90%	165 °C <45 s

**Table 5 sensors-22-05766-t005:** Track width obtained after printing with M5 ink, three layers, and a mesh of 90 threads/cm as a function of orientation. Units: mils.

	T15	T20	T25	T30	T40	T50	T80
0°	19.02 ± 1.05	21.54 ± 1.54	24.23 ± 2.54	31.36 ± 1.56	42.46 ± 1.49	51.00 ± 0.97	79.07 ± 4.31
45°	16.19 ± 3.66	20.08 ± 1.66	25.13 ± 1.74	29.63 ± 3.00	40.33 ± 1.67	48.24 ± 2.90	82.19 ± 3.46
90°	18.05 ± 1.95	21.54 ± 1.40	24.61 ± 1.01	30.98 ± 1.78	42.70 ± 1.51	51.32 ± 1.38	84.90 ± 2.45
315°	17.20 ± 2.10	21.38 ± 1.00	24.21 ± 1.92	31.60 ± 2.12	42.38 ± 2.44	51.81 ± 2.66	82.82 ± 2.75

**Table 6 sensors-22-05766-t006:** Maximum force [N] applied on the *X* axis.

	0805	1206	SOT23
SRA SSLTNC	7.0	4.6	10.5
Chipquick TS3922T50	6.3	4.6	9.0
Indium Indalloy#281	3.7	4.2	6.9
Indium Indalloy#1E	2.9	2.9	6.9

## Data Availability

Not applicable.

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
