# Peer review of "Low-Temperature Soldering of Surface Mount Devices on Screen-Printed Silver Tracks on Fabrics for Flexible Textile Hybrid Electronics"

_sensors, 2022, doi:10.3390/s22155766_

Round 1

Reviewer 1 Report

The authors study the optimal design parameters of screen-printed silver tracks as well as the low-temperature soldering techniques which could be of interest to the researchers working in the field of printed electronics and flexible sensors. The experiment is well designed and elaborated. The authors demonstrated three layers of conductive ink printed with a mesh of 130 threads/cm on a waterproof cotton substrate. The authors also provide some general guidance on textile substrate selection, pretreatment and successful low temperature soldering. I found this work of importance and suggest it be accepted for publication in Sensors.

Reviewer 2 Report

In this paper, the authors mainly reported on the low temperature soldering of surface mount devices on screening-printed silver tracks on fabrics for flexible hybrid electronics. Although the author has done a lot of experimental works, the scientific and logical nature of this article is very weak The whole article reads more like an experimental report, lacking the necessary scientific analysis and conclusions. Below are some specific comments and suggestions, which the referee hopes will be of some help in improving the quality of this paper. 

1. The author describes a lot of content in the Introduction, but the logic is slightly confused, which makes readers miss the main point.  It is suggested that the author further summarize the research progress in related fields, and clearly describe the key and currently existing scientific or technical problems that are mainly solved in this work.  

2. The author has also made some discussions in the experimental part.  The referee suggests the author only describe the specific material used in this work. Descriptions of material properties and others can be placed in the text.

3. When discussing Figure 3, the author should describe in detail what factors determine the “100% Correct Tracks”, and some optical microscope or SEM images should be given to demonstrate the successful printing of tracks, with the best resolution, on fabric. In addition, the author must give some scientific analyses instead of just reporting the results.  

4. The reviewer does not know the purpose of the author to show all the designed pattern in Figure 1.  Not only does it look confusing and misleading, but some of the graphics didn't show up in subsequent research.  

5. Although the author did some shear tests, they were far from sufficient to demonstrate the flexibility of printed materials, integrated devices, or systems.  The author needs to supplement the related experiments to demonstrate the flexibility of electrical tracks and hybrid electronics, such as stretching, bending, twisting, etc.  In addition, the author needs to further analyze the failure mechanism of the presented Flexible Hybrid Electronics. 

Reviewer 3 Report

Dear Editor,

Here are my comments on “Low temperature soldering of surface mount devices on screen-printed silver tracks on fabrics for textile flexible hybrid electronics 

Suggestion: Major revision

Rocío Silvestre et. al investigated screen-printed silver tracks and low-temperature solder on fabrics. The study is systematic and the conclusion is solid. This work definitely can be published, but I think the journal of electronics rather than sensors is more suitable because there is no sensor in this work. Or fabricate a sensor based on this technique as a demo rather than the Oscillator. More suggestions:

1.      The writing should be more concise. There is too much descriptive statement not very closed to the experiment. This type of writing is not read-friendly. Especially for the introduction part. Current introduction is distracting from the significance of this work.

2.      softening (TS), melting (TM) and burning points (TB) are suggested to change as Ts, Tm, Tb

3.      the unit “mils” is suggested to transfer to international standard unit, such as μm

4.      Is Figure 1 one pattern? It is too complicated while Figure 2 is too simple.

Round 2

Reviewer 2 Report

The referee appreciates the explanations and modifications made by the authors, and the manuscript is obviously improved. The referee is also glad to notice that the authors are working with the industry transference of their developed technique. Before the publication in Sensors, however, some minor reversions should be addressed.

1.       Regarding to Comment #3, the referee agrees with the authors that presenting the images of the faults will provide more information to guide the fabrication. The referee also appreciates that a series of optical images of the printed tracks is added at the end of the response letter. Of course not all results need to be published in the paper. Since the authors have optimized the printing details (including the number of printings, mesh size, printing orientations, and others), and successfully obtained minimized tracks (15 mils with 100% correct tracks if the referee was right), a typical optical image is suggested to add in the paper.

2.       The referee supposes that the “correct tracks [%]” for the y-axis of Figure 4 denotes “the percentage of tracks with no errors”. If so, please add the definition of “correct tracks” in the caption of Figure 4.

3.       Please confirm the sentence “a mesh of 105 thread/cm” in L322.

Reviewer 3 Report

For my questions, the authors have solved.

But for Reviewer 2's questions, there still some space for improving. I really appreciate the rigorous attitude of Reviewer 2. Many of his/her comments are keys. For example, "The author describes a lot of content in the Introduction, but the logic is slightly confused, which makes readers miss the main point." Previously, I just simply said "The writing should be more concise." Reviewer 2 made it more accurate. 

From the response letter, the authors did not humbly accept Reviewer 2's suggestions. 
